# Irisin: A Possible Marker of Adipose Tissue Dysfunction in Obesity

**DOI:** 10.3390/ijms241512082

**Published:** 2023-07-28

**Authors:** Laura Tomasello, Maria Pitrone, Valentina Guarnotta, Carla Giordano, Giuseppe Pizzolanti

**Affiliations:** 1Laboratory of Endocrinology and Regenenerative Medicine “Aldo Galluzzo”, Università di Palermo, 90133 Palermo, Italy; maria.pitrone@unipa.it (M.P.); valentina.guarnotta@unipa.it (V.G.); carla.giordano@unipa.it (C.G.); 2Dipartimento Department of Health Promotion, Mother and Child Care, Internal Medicine and Medical Specialties (Promise), Azienda Ospedaliera Universitaria Policlinico Paolo Giaccone, 90127 Palermo, Italy; 3ATeN Center—Advanced Technologies Network Center, 90127 Palermo, Italy

**Keywords:** obesity, ECM remodeling, adipose tissue dysfunction, molecular mechanism, metalloproteinase (MMP), irisin, extracellular matrix, inflammation

## Abstract

Adipose tissue (AT) secretes pro- and anti-inflammatory cytokines involved in AT homeostasis, including tumor necrosis factor-α (TNFα) and irisin. The functionality of AT is based on a regulated equilibrium between adipogenesis and extracellular matrix (ECM) remodeling. We investigated the contributions of adipose progenitors (ASCs) and adipocytes (AMCs) to TNFα-induced ECM remodeling and a possible implication of irisin in AT impairment in obesity. ASCs and AMCs were exposed to TNFα treatment and nuclear factor–kappa (NF-kB) pathway was investigated: Tissue Inhibitor of Metalloproteinase (TIMP-1), Twist Family Transcription Factor 1 (TWIST-1), and peroxisome proliferator-activated receptor-γ (PPARγ) expression levels were analyzed. The proteolytic activity of matrix metalloproteinases (MMPs) -2 and -9 was analyzed by zymography, and the irisin protein content was measured by ELISA. In inflamed AMCs, a TIMP-1/TWIST-1 imbalance leads to a drop in PPARγ. Adipogenesis and lipid storage ability impairment come with local tissue remodeling due to MMP-9 overactivation. In vitro and ex vivo measurements confirm positive correlations among inflammation, adipose secreting irisin levels, and circulating irisin levels in patients with visceral obesity. Our findings identify the NF-kB downstream effectors as molecular initiators of AT dysfunction and suggest irisin as a possible AT damage and obesity predictive factor.

## 1. Introduction

Obesity is the result of multifactorial stimuli and conditions, including chronic low-grade inflammation, oxidative stress, metabolic abnormalities, and immune dysfunction, which represent the principal players in adipose tissue (AT) and its responses [1]. It is accepted that obesity is associated with, and sustained by, a low level of chronic system inflammation, which also reflects what it comes about in AT [2]. When a condition of obesity develops, macrophage infiltration in AT occurs, releasing a massive amount of pro-inflammatory cytokines, including interleukin (IL)-6 and tumor necrosis factor α (TNFα), with consequent induction of several pro-inflammatory pathways and JNK and nuclear factor–kappa B cells (NF-κB) [2,3,4].

During body weight alteration, pre-adipocytes become mature adipocytes and change their size and shape to regulate temporary fat storage in AT [5,6].

Under healthy conditions, the acquisition and maintenance of mature adipose phenotypes are guaranteed by the expression of the peroxisome proliferator-activated receptor gamma (PPARγ) transcription factor [7,8]. In obesity, PPARγ expression is affected, and the biological process is pathologically accelerated to accommodate adipocyte hyperplasia (increase in cell number) and hypertrophy (increase in cell size) [9,10]. Failure to recruit new adipocytes due to the dysregulation of adipogenesis (in both the proliferation and differentiation processes) leads to hypertrophic adipocytes with degradation and deposition of new extracellular matrix (ECM) [9,11].

ECM remodeling is a complex process, involving several proteins with different functions, especially metalloproteinases (MMPs) and their inhibitors (Tissue Inhibitor of Metalloproteinase, TIMPs) and epithelial mesenchymal transition proteins, including Twist Family BHLH Transcription Factor 1 (TWIST-1) [12,13,14]. In tissues, net MMP enzymatic activity is regulated by a balance between the protein levels of activated MMPs and TIMPs [15]. This protein ratio is highly sensitive to the environment: when hostile conditions are established (i.e., inflammation) regulation fails, and adipose tissue dysfunction (ATD) occurs [16,17,18]. MMP-2 and MMP-9 functions have been prominently described in adipogenesis; constitutive MMP-2 and modulated MMP-9 expression were found in adipose cells at different steps of the adipogenesis process and in subcutaneous adipose tissue (SAT) of overweight patients [4,19,20]. At the same time, dysregulation of the balance between TIMP-1 and its co-player, TWIST-1, in obese adipose tissue has been observed [21,22,23]. Independent studies have reported increased circulating levels of TIMP-1 and -2 in patients with metabolic syndrome, diabetes, and obesity. Data concerning their expression level trends in obese adipose tissue remain under debate [24,25].

Although ECM remodeling has been thoroughly investigated, the trigger for the inflammatory network to modify the expression pattern of MMPs/Tissue Inhibitor of MMP system in obesity is unclear. Different intrinsic signals can disrupt the environment, TNFα, and NF-kB cell signaling, the main pathway associated with the inflammatory process [3,4]

Over the past decade, AT has been proposed as an endocrine organ that secretes bioactive peptides, termed ‘adipocytokines’ and has been identified as potential link between obesity and other metabolic disease states, including irisin, which works through autocrine, paracrine, and endocrine effects [26,27,28]. In obese adipose tissue, chronic inflammation leads to the dysfunction of adipocytokines, including irisin, establishing a protective or harmful imbalance [29,30,31]. Previously identified as myokine, irisin has been recognized as an exercise-induced myokine that is able to increase energy, improve glucose tolerance, and reduce insulin resistance [31,32,33,34,35]. Principally, irisin exerts its beneficial effect by shifting mature adipocytes towards a brown phenotype in a biological process defined as the browning of white adipose tissue [36] In mice, the browning effect is well characterized, whereas in humans, irisin has been described in mature adipocytes but not in pre-adipocytes, where it seems able to inhibit adipose terminal differentiation and promote osteogenesis [37,38]. During browning, irisin upregulates the expression of uncoupling protein 1 (UCP1), a protein that generates heat through the catalysis of proton movement across the mitochondrial membrane, causing energy expenditure [39,40]. Zhang et al., suggested that the irisin effects are adipose-stage-dependent differentiation. Irisin treatment, in fact, induces browning in adipocytes, inhibiting the adipogenesis of adipose progenitor cells [41].

The results concerning circulating irisin levels in obese patients are still controversial, especially in the presence of metabolic dysfunction, i.e., dyslipidemia, type 2 diabetes mellitus (T2DM), and other components, which are also characterized by a chronic inflammatory state [42,43,44,45,46,47,48,49]. A meta-analysis conducted by Jia and colleagues showed high circulating irisin levels in obese patients when compared to healthy controls [50]. Furthermore, in vitro and animal studies have attributed anti-inflammatory capabilities to irisin, including the reduction in pro-inflammatory cytokines and the proliferation of macrophages, induction of M2 polarization, inhibition of the inflammasome, and downregulation of NF-kB cell signaling [51].

In our recent study, we indicated a negative correlation between the levels of irisin and IL-6, a pro-inflammatory cytokine involved in metabolic and cardiovascular disorders, in the serum of patients suffering from T2DM under the addition of treatment with glucagon-like peptide 1 (GLP-1) receptor agonists (GLP-1 RA) to any hypoglycemic therapy [49].

Interestingly, irisin, as happens with MMPs, shows a different expression pattern depending on the study model, the stage of maturation, and the tissue location of adipose cells, and these aspects create many difficulties in gaining a full and complete understanding of its intrinsic clinical significance [40,41,52].

In this in vitro study, to identify a possible target, concerning ECM remodeling events to reduce inflammation, we investigated whether chronic TNFα exposure affects adipose progenitor cells and adipocytes differently. Moreover, to clarify the possible role of irisin in ATD, we explored whether a correlation may exist between inflammation, irisin secretion, and adipogenesis. Finally, to propose irisin as a possible predictive marker of obesity, we examined whether circulating irisin levels could be related to the presence of visceral obesity, recognized as the principal risk factor for cardiometabolic disease and T2DM. Our data show that, in ATD, precocious overinduction of NF-kB cell signaling causes a TIMP-1/TWIST-1 imbalance, leading to dramatic ECM remodeling via MMP-9 and a drastic impairment of adipose tissue maturation. Moreover, we found that irisin levels, both locally in adipose tissue and in the sera of patients suffering from obesity, are strictly associated with the grade of inflammation and visceral adiposity.

## 2. Results

### 2.1. Inflammation Impairs Adipose Tissue Function Affecting Adipose Cell Maturation

To determine whether TNFα time-different stimulation induces functional differences between ASCs and AMCs, cell proliferation and the grade of differentiation (as lipid storage capability) were assessed. During the adipose differentiation period, ASCs were kept under TNFα (300 U/mL), mimicking an early and chronically inflamed environment (AMCs_CI_). Untreated ASCs and AMCs were used as negative and positive controls, respectively. Optical microscopy observations showed a lower cell density and dramatic decrement in the droplet accumulation capability in AMCs_CI_ when compared to untreated AMCs (Figure 1a–c). The relative quantitative analysis of the oil-red-stained area percentage (Figure 1d) revealed a lower lipid storage capability in AMCs_CI_ compared to untreated AMCs (6.53 ± 0.82% vs. 72.88 ± 4.49%, *p* value < 0.001). Cell cycle analysis revealed different cell cycle distributions. A physiological decrement in the proliferation index (PI) in AMC vs. ASCs was measured (PI 0.24). Moreover, a significant decrement in proliferation was found in AMCs_CI_ when compared both to ASCs and untreated AMCs, respectively up to 50.08 ± 2.18% (Figure 1e) and up to 68.24 ± 1.87% (*p* < 0.005).

### 2.2. TNFα Induces the NF-kB Pathway in AMCs

As NF-kB cell signaling represents the main TNFα responsive upstream factor, its activation was evaluated in ASCs and AMCs after pro-inflammatory treatment. Proteins were extracted from untreated and TNFα (300 U/mL)-treated ASCs up to 72 h and in AMCs_CI_. ASC52telo and subcutaneous adipose tissue from patients suffering from obesity (referred to as SAT) were used as controls. Western blot (WB) assays were performed, and the relative optical density was measured (Figure 2, right panel). Specifically, a modest NF-kB activity was highlighted (Figure 2, right panel). In detail, we found a significant induction of NF-kB fc in untreated AMCS and AMCs_TNFα72 h_ compared to the control ASCs (1.92 ± 0.24 fc and 2.72 ± 0.35 vs. ASCs *p* < 0.001). There were significant increments in AMCs_TNFα72 h_ and AMCs_CI_ (Figure 2b) when compared to untreated AMCs (1.40 ± 0.15 fc and 1.67 ± 0.19 fc vs. AMCs, *p* < 0.001). Significant NFk-B hyperactivation was detected in AMCs_CI_ vs. AMCs (about 52% up in AMCs_CI_ vs. AMCs, *p* < 0.001). There were no differences in AMCs_CI_ and AMCs_TNFα72 h_ (*p* > 0.05).

These data firstly suggest that the activation of NF-kB cell signaling represents an obligatory event in adipose differentiation and maturation processes and, secondly, indicate that differentiated adipocytes from SAT of obese patients have high basal levels of NF-kB. It is reasonable to speculate that TNFα prematurely induces NF-kB in pre-adipocytes, compromising their full maturation.

### 2.3. TNF-α Is Crucial for Adipose Maturation

As NF-kB cell signaling is well-known to be involved in MMP activation, the mRNA expression levels of NF-kB, MMP-9, and MMP-2 and of their principal regulators, TIMP-1 and TWIST-1, were analyzed. ASCs and AMCs were treated with 300 U/mL TNFα for 72 h and compared to AMCs_TNF-CI_ through a qRT-PCR experiment. In detail, a significant increment in NF-kB mRNA expression levels was found in AMCs compared to ASCs (3.42 ± 0.31-fc vs. ASCs), *p* < 0.001). Significant differences were found in AMCs_TNFα(72 h)_ and in AMCs_CI_ with upregulation of NF-Kb by approximately 36.8% and 29.2%, respectively, when compared to AMCs. Moreover, AMCs_TNFα(72 h)_ NF-kB mRNA expression levels showed no difference compared to AMCs_CI_, whereas upregulation of 1.13 ± 0.11 and 3.75 ± 0.27-fold was detected compared to AMCs and ASCs, respectively. AMCs showed the overexpression of MMP-9 and MMP-2 by 2.23 ± 0.18-fc and 18.23 ± 0.2-fc compared to ASCs. AMCs_CI_ and AMCs_TNFα(72 h)_ exhibited higher mRNA expression levels of MMP-9 and MMP-2 (3.46 ± 0.75-fc and 3.25 ± 0.23-fc, 25.06 ± 0.42-fc and 25.25 ± 0.14-fc vs. ASCs, respectively). Interestingly, no impacts of TNFα treatment on ASCs and no significant differences in AMCs_CI_ and AMCs_TNFα(72 h_) were detected (Figure 3b,c). TIMP-1 and TWIST-1 showed opposite trends in expression. We found decrements of approximately 43.30% and 48.7% and 44.42%, and 27.15%, respectively, for TIMP-1 and TWIST-1 expression in AMCs_CI_ and AMCs_TNFα(72 h)_ when compared to AMCs (Figure 3a–e).

Finally, the mRNA expression levels of the key adipose gene specific marker, PPARγ, were downregulated in AMCs_TNF-CI._ No difference was observed in AMCs_TNFα(72 h)_ compared to AMCs, whereas there was a significant downregulation of approximately 40.42% in AMCs_TNF-CI_ compared to AMCs_TNFα(72 h)_, suggesting that inflammation affects early adipose terminal differentiation (Figure 3f).

### 2.4. Chronic Inflammation Results in Local Tissue Remodeling

To evaluate the enzymatic activity of MMP-2 and MMP-9, gelatin substrate gel zymography was assessed. Different activity patterns for ASCs were detected for MMP-2 and MMP-9 (Figure 4a). No TNFα inducible activity was detected in ASCs and ASCtelo (used as positive controls), which showed no MMP-9 and, in contrast, constitutive expression of MMP-2. In contrast, in adipocytes (derived from obese SAT), the modulation of MMP-9, in both pro and active forms, was appreciable. TNFα significantly modulated the proteolytic function of MMP-9, inducing increments of approximately 0.57-fc and 0.72-fc, respectively, in AMCs_CI_ and AMCs_TNFα(72 h)_ when compared to AMCs (Figure 4b). These data suggest that, in the absence of stimuli, ECM remodeling is not a persistent event.

### 2.5. Mature Adipocytes Release Irisin in a TNFα Dose- and Time- Dependent Manner

As altered irisin levels were previously found in serum of T2D patients under additive treatment with GLP-1 RA, we investigated whether a correlation between adipose irisin release and inflammation exists [49]. Firstly, the protein contents secreted in SATob treated with several concentrations of TNFα (200, 300 and 500 U/mL) at 24 and 48 h and in AMCs_CI_ exposed to the same TNFα concentrations up to 72 h were measured. Finally, we coupled adipose-secreting irisin protein levels with PPARγ mRNA level expression and circulating irisin levels with IL-6 and visceral obesity. We found values of 1.38-fc and 2.77-fc in SATob treated with 200 U/mL of TNFα, 1.61-fc and 4.25-fc in SATob treated with 300 U/mL of TNFα, and 2.93-fc and 5.85-fc in SATob treated with 500 U/mL of TNFα when compared to AMCs at 48 h and 72 h, respectively (*p* < 0.05) (Figure 5a). We found lower protein contents secreted in AMCs_CI,_ at 200, 300 and 500 U/mL TNFα when compared to untreated AMCs, of about 0.69-fc, 0.87-fc, 1.03-fc respectively (*p* < 0.05). TNFα dose-dependent irisin relative release appeared to be weakly present. In line with several studies, higher secreted irisin levels were detected in ASCs (3.27 ± 0.15 µmL). Increments of about 26% and 18%, respectively, were detected in AMCs + 500 U/mL TNFα and AMCs_CI_, + 300 U/mL TNFα when compared to AMCs_CI_, + 200 U/mL TNFα (Figure 5b).

To investigate whether irisin might be a marker of maturation and the adipose function grade, a correlation analysis between the adipose-secreted level of irisin and the gene expression level of PPARγ was carried out. We found that PPARγ expression levels are negatively related to irisin released by AMCs when treated with TNFα. In detail, dose- and time-dependent relations to TNFα treatment were detected: decrements of 57.13 ± 1.81% and 54.63 ± 4.12% in adipose irisin release were accompanied by lower PPARγ mRNA expression levels in AMCs exposed to 500 U/mL TNFα at 48 and 72 h when compared to untreated AMCs (3.71 ± 0.2 vs. 6.21 ± 0.31-fc and 3.15 ± 0.14 vs. 6.35 ± 0.21-fc) at 48 and 72 h, respectively. Figure 5c–e).

Obesity is characterized by prolonged low-grade tissue and systemic inflammation mediated by high serum concentrations of circulating pro-inflammatory cytokines, including IL-6. To investigate whether irisin might be a predictive marker of disease, the circulating basal levels of irisin were compared to circulating basal levels of IL-6 and the waist circumference.

The analysis revealed that irisin is positively related to IL-6 circulating levels and visceral obesity (13.9 ± 0.89 ng/mL and 5.26 ± 1.83 pg/mL vs. 11.07 ± and 3.69 ± 2.38, in patients with visceral obesity vs. patients without visceral obesity. Figure 5e,f).

Together, these results suggest that a signaling mechanism linking inflammation and irisin expression could be hypothesized.

## 3. Discussion

Obesity involves 170 million children (under the age of 18) and 650 million adults, with growing prevalence in low- and middle-income countries [52]. Obesity complications include type 2 diabetes mellitus (T2DM), thyroid diseases, hypertension, cancer, and depression, representing a worldwide concern both for the negative impact on the quality of life and for the associated elevated annual health-care costs [53]. 80% of patients suffering from T2DM also suffering from obesity, whereas 10–30% of people with obesity show no metabolic alteration [54]. of patients suffering from T2DM also suffering from obesity, whereas 10–30% of people with obesity show no metabolic alteration [54]. Although obesity is related to an excess weight gain (expressed as an increase of the body mass index (BMI)), it is reductive to define it as an energy imbalance between calorie intake and expenditure. The established link between inflammation and development of insulin resistance (IR) and T2DM is well recognized. Inflammation, resulting from the progressive development of obesity, may play a role in inducing IR [3,55]. However, an impairment of AT remodeling in obesity is not necessarily found, and IR does not necessarily occur in obesity. Inflammation could also be driven by adipose tissue expansion through the activation of several cell activation signals due to interactions between the cells and the extracellular matrix (ECM). Independent studies show that tumor necrosis factor alpha (TNFα) levels are essentially dependent on the visceral fat amount and are positively associated with higher values of glycated hemoglobin (HbA1c) in patients with T2DM [56,57,58]. Conversely, the cellular and molecular contributions of affected adipose tissue (AT) are not well-described. During fat mass expansion, extracellular matrix (ECM) remodeling with the commitment of metalloproteinases occurs (MMPs) [4,19,59]. MMPs, a family of zinc-dependent endopeptidases, are primarily involved in enzymatic degradation of the ECM, and their function is central to many physiological and pathological processes, including adipogenesis [60,61]. MMPs are released in an inactive enzymatic form (pro-MMPs) and become active in the extracellular environment once they have been cleaved. In several chronic inflammation diseases, it has been observed that MMP activity is also affected by the balance between TIMP-1 and TWIST-1, but the effect on adipogenesis has not been clarified [62]. Moreover, embryogenesis in vivo studies performed in both *Drosophila* and mice revealed an adipose deficiency and atrophy in multiple tissues from mutant animals, suggesting negative feedback regulation by TWIST-1 that represses the NF-kB-dependent cytokine pathway [62,63]. Although MMP involvement and ECM remodeling in the adipose differentiation process are well-established, their regulation in obese adipose tissue dysfunction is controversial [16,17].

A crucial event in physiological adipose tissue remodeling is pre-adipocyte recruitment and commitment by peroxisome proliferator-activated receptor gamma (PPARγ), a key regulator of adipogenesis [64]. Dysregulation of PPARγ can disrupt the balance between adipocyte differentiation and lipid storage, leading to hypertrophy (accumulation of triglycerides in enlarged adipocytes) and can alter the production and release of adipocytokines, leading to an imbalance between energy homeostasis and anti- and pro-inflammatory factors, irrevocably contributing to the impairment of AT functionality, a condition known as AT dysfunction (ATD) [9,11]. Among the adipocytokines described in AT homeostasis, irisin is arousing great interest. Despite its widely described potential beneficial role for browning white adipose tissue and inducing thermogenesis gene expression, the therapeutic potential of irisin is controversial [65,66,67]. The discrepancy in the results principally regards its involvement in the inhibition of the adipogenesis process by downregulating PPARγ expression via the induction of Wnt/β–catenin signaling and the divergence between evidence observed in animal models and in humans [37,38,51]. Recent studies have indicated that irisin is a predictive marker of several pathologies, including sarcopenia, atherosclerosis, and heart failure [45,68,69,70].

In this in vitro study, we aimed to clarify whether the different statuses of inflammation affect mature adipocytes (AMCs) and their progenitors, adipose mesenchymal stem cells (ASCs), in a different dose- and time-dependent manner, and if irisin could be recognized as a possible predictive marker of obesity. For this purpose, we investigated the effect of TNFα at different concentrations, mimicking the different grades and moments of inflammation in obese adipose tissue in ASCs, AMCs, and chronically inflamed AMCs. We focused on NF-kB cell signaling, the master pathway induced by TNFα, and its downstream mediators, including MMPs, especially MMP-2 and MMP-9, and their enzymatic activity modulators, TIMP-1 and TWIST-1. Our data reveal that different TNFα treatment conditions evoke different responses in AMCs and ASCs. Chronically inflamed AMCs progressively lose the terminal adipose phenotypic features: lipid droplet accumulation capability and mRNA expression of the adipose specific marker, PPARγ, dramatically fall, whereas irisin release increases. Based on the comparative analysis between SAT derived from obese patients, pre-adipocyte progenitor cells, and mature adipocytes, we speculate that the premature activation of NF-kB cell signaling leads to pathological ECM remodeling through the proteolytic activity of MMPs (primarily of MMP-9) and alteration of the adipocyte maturation grade. Although NF-kB cell signaling induction obligatorily occurs during the adipose maturation process, we found that its premature overactivation causes dysregulation of the MMPs/TIMP-1 and TIMP-1/TWIST-1 balance, probably preventing the expression of PPARγ [71]. Moreover, our findings suggest that PPARγ expression levels could also be disturbed by increased adipose irisin secretion. The irisin secretion trend observed was accompanied by an opposite trend in PPARγ expression levels: pre-adipocytes released large amounts of irisin and did not express PPARγ, and weak irisin secretion was detected in adipocytes with higher PPARγ expression levels, whereas adipocytes showed significantly increased irisin secretion when exposed to a rising concentration of TNFα, while a drastic downregulation of PPARγ was detected. The irisin adipose secretion movement is in line with increasing circulating irisin levels detected in the serum of patients with visceral obesity and a higher systemic inflammation basal grade. Due to the negative correlations found between adipose irisin release, inflammation, the occurrence of visceral obesity, the proliferation index, and the impairment of the lipidic storage capability in inflamed adipocytes, we speculate that adipocytes release irisin to counteract the effects of inflammation and mass fat expansion.

## 4. Materials and Methods

### 4.1. Ethical Statement

The protocol was approved by the Independent Ethical Committee (no. 08/2018; 27 August 2018) at the Azienda Ospedaliera Universitaria Policlinico “Paolo Giaccone”, Palermo, Italy. All patients gave their written informed consent.

#### Population

Obesity was defined by a body mass index (BMI) ≥ 25 kg/m^2^; visceral obesity was defined by a waist circumference > 102 cm in males or 88 cm in females. Seventy-one patients suffering from obesity referred to the Division of Endocrinology of the University of Palermo from March 2019 to May 2020 were consecutively included in the correlation analysis. Among them, there were 40 with visceral obesity and (56%) and 31 without (44%) visceral obesity. Inclusion criteria were glycated hemoglobin (HbA1c) < 6.5% (48 mmol/mol), 20–70 years in age, and BMI ≥ 25 kg/m^2^. Exclusion criteria were the presence of diabetes, systemic inflammatory disease, pregnancy, infectious disease, a personal history of cancer or multiple endocrine neoplasia type 2, and acute or chronic liver injury defined as the elevation of transaminases or bilirubin or alkaline phosphatase ≥ 3 above the upper limit of the normal value.

### 4.2. Human Biological Sample

#### 4.2.1. Subcutaneous Adipose Tissue (SAT)

Subcutaneous (SAT) adipose tissue biopsies were obtained from seventy-one consenting patients (BMI) ≥ 25 kg/m^2^): 40 obese and 31 non-obese subjects undergoing elective open-abdominal and laparoscopy surgery.

#### 4.2.2. Cell Culture: Adipocyte, Adipose Mesenchymal Stem Cells (ASCs), and Adipose Differentiation

The stromal vascular fraction (SVF) and adipocytes were obtained as previously described [72]. Briefly, bioptic specimens kept in DMEM/Ham’s F12 1:1 were mechanically dissected from fibrous material and visible blood vessels, cut into little fragments, and incubated in PBS Ca^2++^/Mg^++^ (phosphate-buffered saline with calcium and magnesium) (Sigma Chemical, St. Louis, MO, USA) supplemented with 1 mg/mL collagenase type I (Sigma Chemical, St. Louis, MO, USA) with vigorous shaking (100 cycles/min) for 1 h at 37 °C. The samples were filtered and centrifugated to separate adipocytes and free oil from the SVF (presenting the ASCs).

ASC expansion: After isolation, ASCs were cultured in the complete culture medium, consisting of Dulbecco’s Modified Eagle Medium (DMEM) and Ham’s F12 with L-glutamine (Euroclone S.p.a, Pero, MI, Italy) supplemented with 5% fetal bovine serum (FBS, Euroclone S.p.a, Pero, MI, Italy), and passaged when 80–90% confluence was reached [72]. ASCs between culture passages four and seven were used in the experiments.

Differentiation protocol: For adipose differentiation, ASCs were cultured in an adipose differentiation medium consisting of DMEM:F12 with 500 µmol/L of 3-isobutyl-1-methylxanthine (IBMX, Gibco, Gaithersburg, MD, USA), 10^−4^ mM of dexamethasone (Sigma-Aldrich, Merck KGaA, Darmstadt, Germany), 100 μM of indomethacin (Sigma-Aldrich, Merck KGaA, Darmstadt, Germany), and 1 µg/mL of insulin at 37 °C in 5% CO_2_ for up to 21 days. Once the differentiation period had been completed, lipid droplets were detected by oil red O staining (Sigma-Aldrich, Merck KGaA, Darmstadt, Germany) according to the manufacturer’s procedures and observed by optical microscopy [72].

Adipocytes:

SATob: adipocytes were isolated from the bioptic SAT of patients suffering from obesity by centrifugation and filtration, as previously described [72].

AMCs_CI_: differentiated AMCs derived from ASCs were exposed to 300 U/mL TNFα treatment during the differentiation period, which we refer to as chronically inflamed AMCs (AMCs_CI_)

Positive control cell line: the ASC52telo (SCRC-4000, ATCC, Manassas, VA, USA) hTERT immortalized adipose-derived mesenchymal stem cell line.

#### 4.2.3. Serum Sample for the Correlation of Visceral Obesity with IL-6 and Irisin

##### Laboratory Assays

Venous blood was collected from each patient into sterile 5 mL vacutainer serum separator tubes with clot activator (SST; Becton Dickinson, Franklin Lakes, NJ, USA). Serum samples were assayed for irisin and IL-6 concentrations using a commercial kit (respectively, EK-067-29; Phoenix Pharmaceuticals, Karlsruhe, Germany, and electrochemiluminescence assay, Roche, Milan, Italy) following the manufacturers’ instructions.

### 4.3. Cell Treatment: ASC TNFα Stimulation

For the stimulation experiments ASCs52Telo, ASCs, AMCs, and SATob were seeded at a cell density of 2 × 10^4^/cm^2^ in six-well cell culture plates. The day after, they were stimulated with 200, 300, or 500 U/mL TNFα (PeproTech, Inc., London, UK) for up to 48 and 72 h.

Low-chronic-inflammation-mimicking conditions: Differentiated AMCs were kept under 300 U/mL TNFα for a 21-day differentiation period (referred as to AMCs_CI_).

Untreated cells were used as controls. (1 U = 2.7 × 10^5^ according to R&D Systems).

### 4.4. Flow Cytometry Analysis

Single-cell suspensions of ASCs, AMCs, and AMCs_CI_ were obtained, and a DNA content analysis was performed according to Nicoletti’s protocol. Briefly, 1 × 10^6^ cells were fixed in 70% ethanol, rehydrated in phosphate-buffered saline (PBS), and then resuspended in a DNA extraction buffer (with 0.2 M NaHPO_4_ and 0.1% Tritonx-100 at pH 7.8). After staining with 1 μg/mL propidium iodide for 5 min, the fluorescence intensity was determined by analysis on a FACS Calibur flow cytometer (Becton-Dickinson, East Rutherford, NJ, USA). Data acquisition was performed with CellQuest (Becton Dickinson) software, and the percentages of G1, S, and G2 phase cells were calculated with the MODFIT-LT 2.0 software program (Verity Software House, Inc., Topsham, ME, USA). The proliferation index (PI) was measured by the following equation [73]:$$\mathrm{PI}=\frac{phase\;G2\;num\;cells}{\left(phase\;M\;num\;cells+phase\;S\;num\;cells\right)}$$

### 4.5. Gene Expression: Quantitative Real-Time-PCR (qRT-PCR)

Total RNA was extracted from ASCs with or without 300 U/mL TNFα for up to 72 h, AMCs with or without 300 U/mL TNFα for up to 72 h, and AMCs_CI_ by the RNeasy kit (Qiagen, Hamburg, Germany), according to the manufacturer’s instructions. The amplification reaction of genes was performed using the Quantitect SYBR Green PCR Kit (cod. 204243, Qiagen, Hamburg, Germany) on the RotorGene Q Instrument (Qiagen, Hamburg, Germany). The amplification of specific genes was confirmed by the melting curve profiles at the end of each qRT-PCR. Relative expression levels for each gene were assessed using the 2^−∆∆Ct^ method by normalization for β-actin. The results were presented as histograms with GraphPad Prism 5 software (GraphPad Software, Inc., La Jolla, CA, USA). The primers used for qRT-PCR reactions are listed in Table 1.

### 4.6. MMP Activity Evaluation: Zymography

AMCs were treated with 300 U/mL TNFα. After treatment, the cells were centrifuged, and condition media were collected. The protein concentration was evaluated by the Bradford assay (Bio-Rad Laboratories S.r.l., Segrate, MI, Italy). A total of 40 µg of the non-reduced protein sample was loaded on 7.5% SDS-polyacrylamide gels containing 1 mg/mL gelatin (Sigma-Aldrich, Merck KGaA, Darmstadt, Germany), following the Gelatin Zymography Protocol developed by Abcam (https://www.abcam.com/protocols/gelatin-zymography-protocol, accessed on 24 July 2023). Gels were stained with 0.5% Coomassie Blue R-250 (Sigma-Aldrich, Merck KGaA, Darmstadt, Germany) and destained in 10% acetic acid. Enzymatic activities appear as clear bands on a dark background. Equal amounts of cell lysate were loaded for the gelatin zymography test. The data obtained by densitometry for each band were referred to as the standard protein and expressed as relative peaks of the area by ImageJ software V1.52. The results were presented as histograms with GraphPad Prism 5 software (GraphPad Software, Inc., La Jolla, CA, USA).

### 4.7. Cell Signaling Investigation: Western Blotting

ASCs and AMCs kept under experimental conditions (72 h and 21 days TNFα 300 U/mL) were scraped and incubated in ice for 30 min with RIPA buffer (50 mM Tris-HCl, pH 7.4, 150 mM NaCl, 1% Nonidet P40) and a protease inhibitor cocktail (Complete EDTA-free, Roche Molecular Biochemicals, Merck KGaA, Darmstadt, Germany). The total cellular lysate was centrifuged at 14,000 rpm for 1 h to clear cell debris. The protein concentration was determined using the Bradford assay. Proteins were denatured in Laemmli sample buffer (2% SDS, 10% glycerol, 5% 2-mercaptoethanol, 62.5 mM Tris-HCl pH 6.8, 0.004% bromophenol blue), separated on 12% polyacrylamide gels, transferred to nitrocellulose membranes (TransBlot Transfer Medium Bio-Rad Laboratories S.r.l., Segrate, MI, Italy), and blotted with the primary antibodies listed in Table 2. Antigen–antibody complexes were visualized using the SuperSignal West Femto Maximum Sensitivity Substrate (Pierce) on a CCD camera (Chemidoc, Bio-Rad Laboratories S.r.l., Segrate, MI, Italy).

The ASC52telo (SCRC-4000, ATCC, Manassas, VA, USA) hTERT immortalized adipose derived mesenchymal stem cell line was used as a positive control. Western blot bands were quantified by densitometry using ImageJ software V1.52 and the results were presented as histograms using GraphPad Prism 5 software (California).

### 4.8. Irisin Production and Secretion: ELISA Assay

Mature adipocytes from SAT from obese patients were exposed to three different TNFα concentrations, 200, 300, and 500 U/mL. Irisin release at 48 and 72 h was evaluated by ELISA assay (irisin, recombinant human, mouse, rat, canine ELISA protocol, Phoenix Pharmaceuticals, Inc., Burlingame, CA, USA), according to the manufacturer’s instructions.

The relative irisin release was evaluated in adipose-differentiated ASCs kept under different TNFα concentrations (200, 300, and 500 U/mL). The results were presented as histograms with GraphPad Prism 5 Software (GraphPad Software, Inc., La Jolla, CA, USA).

### 4.9. Protein Interaction Network and Pathway Representation

Network analysis was performed on the modulated genes coding for the invariant and variant proteins using the STRING (Search Tool for the Retrieval of Interacting Genes/Proteins) website (http://stringdb.org/, accessed on 1 January 2023). The suggested pathway was presented as a flowchart using SmartDraw Software, LLC (free on https://www.smartdraw.com/, accessed on 1 April 2023).

### 4.10. Data Analysis and Statistics

Data are presented as the mean ± SD. All analyses were performed using GraphPad Prism 5 Software (GraphPad Software, Inc., La Jolla, CA, USA). The statistical analysis was performed with an unpaired *T*-test or ANOVA test and a Tukey’s post-analysis. *p*-values ≤ 0.05 were considered statistically significant. The analysis correlation was performed with SPSS version 19 (SPSS, Inc., Chicago, IL, USA). Data are presented as the mean and ± SD. The differences between the two groups were evaluated with an unpaired *t*-test and the Shapiro–Wilk test. Bivariate correlation analyses were used to identify relationships between changes in the variables. A *p* value ≤ 0.05 was considered statistically significant.

## 5. Conclusions

In conclusion, we speculate that in SAT of patients suffering from obesity, inflammation leads progressively to AT dysfunction from the earliest stage of adipogenesis differentiation. TNFα leads to a TIMP-1/TWIST-1 imbalance via the NF-kB pathway, causing downregulation of the expression of PPARγ, the main adipose-specific marker, and TIMP-1/MMPs imbalance with the overactivation of MMP-9, the principal cytoskeletal remodeling protein (Figure 6b).

It is important to mention that this study did not aim to identify TNFα as the only pro-inflammatory cytokine responsible for adipose tissue dysfunction. Many other cytokines contribute to the establishment of the chronically inflamed environment. Our results add knowledge about the different behaviors of mature adipocytes and their progenitors in response to a pro-inflamed environment, highlighting different cellular and molecular contributions to ATD. These findings confirm the importance of acting promptly to address inflammation and that of two interesting players: TWIST-1 and irisin.

However, there are several limitations to our study. Further silencing experiments are needed to clarify the role of TWIST-1 as a key regulator of balance in ECM remodeling and a possible anti-obesity target. Secondly, although the role of irisin as a driver of the browning of white adipose tissue is largely consistent with the literature, we did not investigate this aspect. Additionally, we did not examine irisin as a possible pharmacological target but suggest that irisin is a possible predictive marker of obesity related to systemic and adipose tissue inflammation. Finally, the small sample size may limit the generalizability of the results. However, our preliminary data could be considered significant as a basis for future wider population analyses.

## Figures and Tables

**Figure 1 ijms-24-12082-f001:**
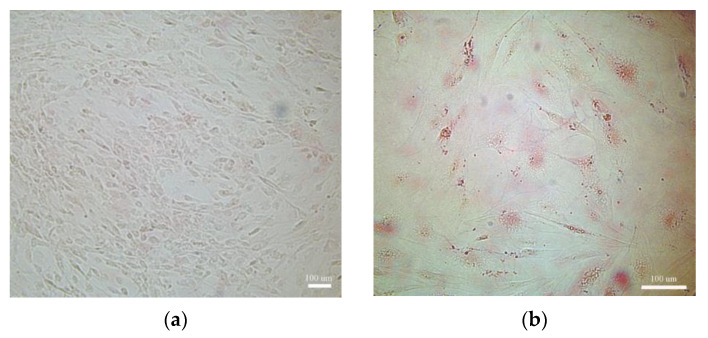
TNFα affects functional maturation and proliferation. (**a**–**c**) Oil red staining. A representative optical microscopy image of undifferentiated ASCs (negative controls), AMCs after 21 days of TNFα treatment (AMCs_CI_). (**d**) The histogram graph represents quantification of the red area (lipid deposits stained with oil red) by ImageJ. (**e**) The histogram graph represents the comparative cell cycle distribution analysis between ASCs, AMCs_CI_, and 21-day AMCs. CI: 21 days of TNFα treatment. The data are expressed as the mean of three different measurements, ±SD. *** *p* < 0.001.

**Figure 2 ijms-24-12082-f002:**
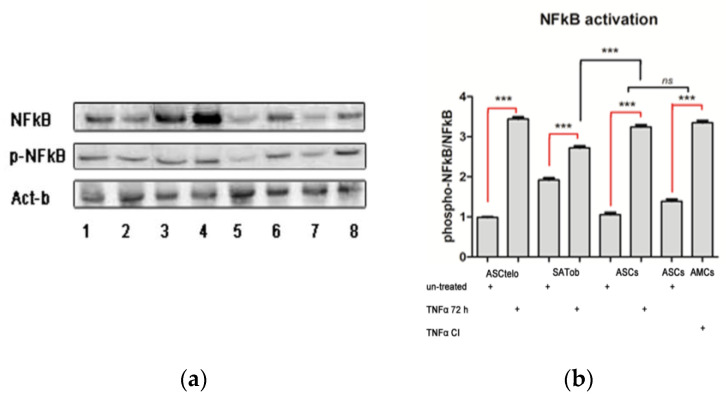
NF-kB activation via TNFα treatment affects adipose differentiation. (**a**) Western blot (WB) analysis: a representative WB assay is presented in the left panel, showing gel bands consistent with p65 (NF-kB), phospho(p)-p65 (pNF-kB), and Act-b (β-actin, normalized protein): ASC52Telo (line 1), ASC52Telo + TNFα (72 h) (line 2); SATob (line 3), SATob_TNFα (72 h)_ (line 4), ASCs (line 5), AMCs + TNFα_CI_ (line 6), ASCs (line 7), and ASCs + TNFα (72 h). (**b**) The histogram graph represents the p-NF-kB and NF-kB quantitative protein ratio based on optical density values and normalized on CP β-actin (**right** panel). CP: ASC52telo positive controls; CI: 21 days of TNFα treatment. The data are expressed as the mean of three different measurements, ±SD. *** *p* < 0.001, and *ns p* > 0.05.

**Figure 3 ijms-24-12082-f003:**
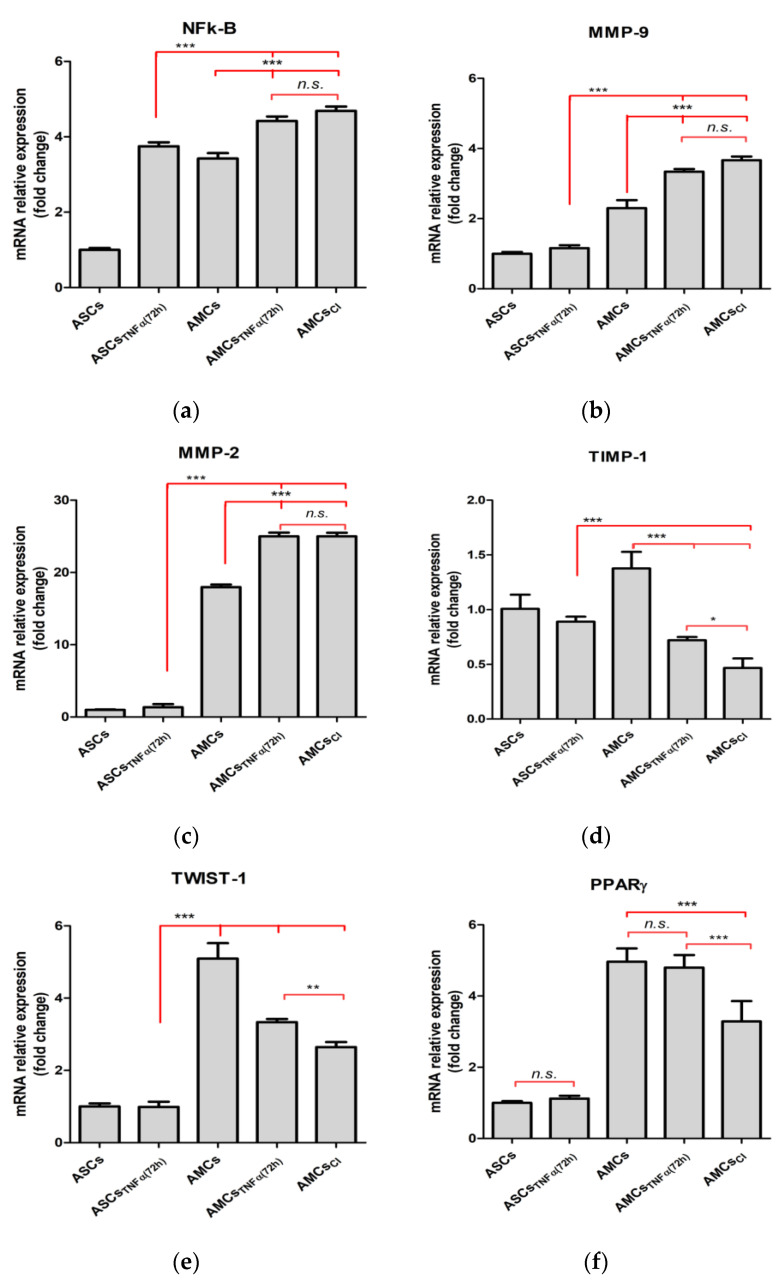
TNFα negatively affects the specific adipose gene profile. The histogram graphs show gene expression levels of (**a**) NF-kB; (**b**) MMP-9; (**c**) MMP-2; (**d**) TIMP-1; (**e**) TWIST-1; and (**f**), PPARγ in ASCs, ASCs + TNFα (72 h), AMCs, AMCs + TNFα (72 h), and AMCs_CI_. CI: 21 days of TNFα treatment. The data are expressed as the mean of three different measurements, ±SD. * *p* < 0.05, ** *p* < 0.005, *** *p* < 0.001, and *n.s. p* > 0.05.

**Figure 4 ijms-24-12082-f004:**
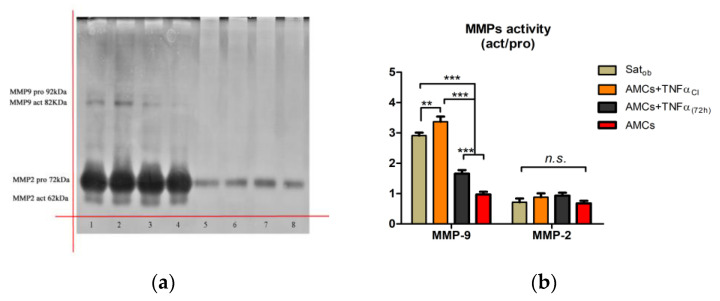
TNFα affects MMP activity. (**a**) Representative zymography gel (inverted color) of the effects of TNF-α on MMP-2 and MMP-9 release: SATobese, adipocytes derived from subcutaneous adipose tissue of patient suffering from obesity (line 1); AMCs (line 2); AMCs_CI_ (line 3); AMCs_TNF-α (72 h)_ (line 4); ASCs (line 5), ASCs _TNF-α (72 h)_ (line 6); ASCsTelo (line 7); and ASCsTelo _TNF-α (72 h)_ (line 8). kDa (KiloDalton), pro (pro-enzyme), act (active form), MMP (metalloproteinase). (**b**) The histogram graph represents MMP-2 and MMP-9 activity expressed by fold change. CI: 21 days of TNFα treatment. The data are expressed as the mean of three different measurements, ±SD. ** *p* < 0.005, *** *p* < 0.001, and *n.s. p* > 0.05.

**Figure 5 ijms-24-12082-f005:**
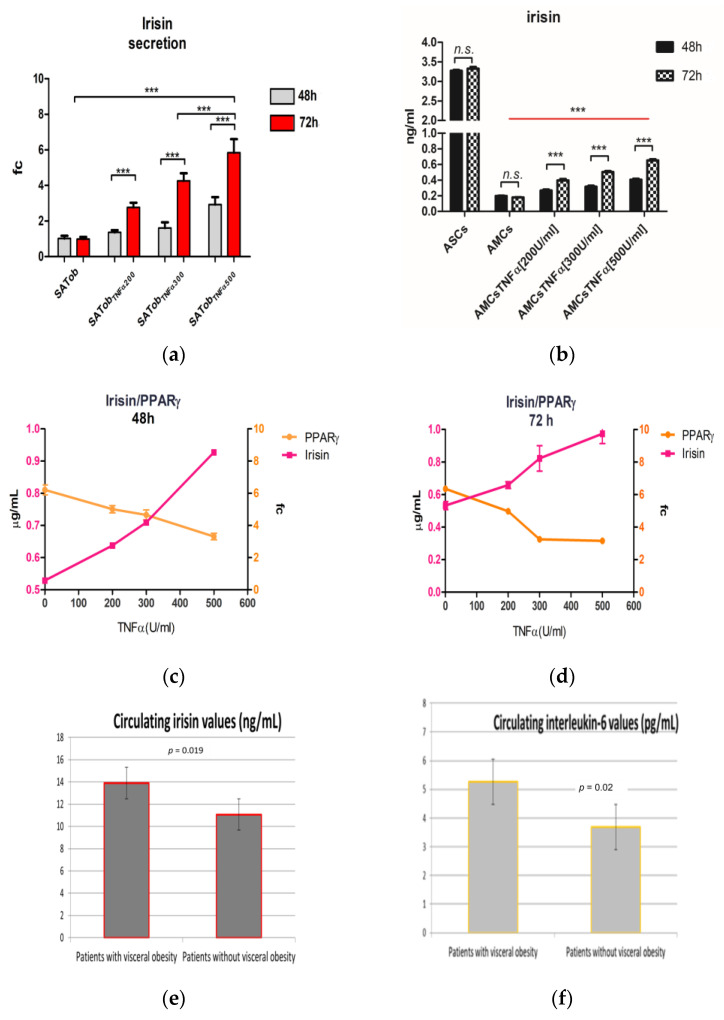
Irisin secretion. (**a**) Comparative irisin secretion by adipocytes derived from subcutaneous adipose tissue of patients suffering from obesity (SATob) exposed to different concentrations of TNFα at 48 and 72 h. (**b**) Comparative relative irisin secretion in AMCs and AMCs exposed to different concentrations of TNFα at 72 h. (**c**,**d**) The line graphs represent the correlation analysis of irisin (protein secreted levels) and PPARγ (mRNA expression levels) related to TNF-α treatment at 48 (**c**) and 72 h (**d**). (**e**,**f**) The bar graphs represent the correlation analysis between the visceral obesity and basal serum levels of irisin and IL-6 (**e**, **f**, respectively). The histogram and line graph represent three different sets of experiments. The data are expressed as the mean ± SD; *** *p* < 0.0001, *n.s. p* > 0.05.

**Figure 6 ijms-24-12082-f006:**
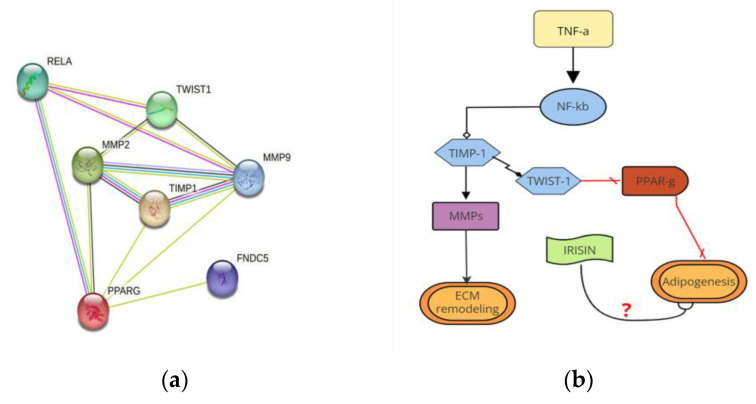
Protein interaction network and pathway representation. (**a**) The node graph represents an analysis of the protein interaction network analysis that occurs in AMCs, highlighting inflammation, ECM remodeling, and possible irisin involvement; (**b**) Flow diagram of the proposed molecular mechanism correlating TIMP-1 with ECM remodeling, dysregulation, and adipogenesis failure in AT dysfunction; (**c**) The flowchart summarizes the responses of pre-adipocytes and adipocytes to TNFα in healthy and obese patients. The difference in circulating irisin levels in patients suffering from obesity with or without visceral obesity is highlighted.

**Table 1 ijms-24-12082-t001:** Primer sequences.

Gene	Sequence (5′-3′)/Code	
MMP-2	QT00088396	Qiagen
MMP-9	QT00011956	Qiagen
NF-Kb	F: GCAGGTTGTTCTGGAAGTTG	MWG
	R: CTGGGGTTTTTCCCTCTCTT	
TWIST-1	F: GTCCGCAGTCTTACGAGGAG	MWG
	R: CTTGAGGGTCTGAATCGGGCT	
TIMP-1	F: CTGTTGTTGCTGTGGCTGATA	MWG
	R: CCGTCCACAAGCAAGAGT	
PPARγ	F: GAGTTCATGCTTGTGAAGGATGC	MWG
	R: CGATATCACTGGAGATCTCCGCC	
β-actin	QT00095431	Qiagen

**Table 2 ijms-24-12082-t002:** Primary Antibody codes.

Protein	Primary Antibody Code	
NF-kB p65	51-0500	Invitrogen
Phospho-p65	14-9864-82	Invitrogen
β-actin	15G5A11/E2	Invitrogen

## Data Availability

Data are contained within the article.

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
