# Peer review of "Irisin: A Possible Marker of Adipose Tissue Dysfunction in Obesity"

_ijms, 2023, doi:10.3390/ijms241512082_

Round 1
Reviewer 1 Report
This article discussed extracellular matrix remodeling induced by TNF-α as an early key event in adipose tissue dysfunction via NF-kB cell signaling in an MMP-dependent manner. There are some concerns in this manuscript that should be addressed as follows:
1. Title: "Irisin" should be mentioned in the title.
2. Page 1 Line 18: The meaning of the abbreviations should be explained at their first mention; e.g. " TIMP-1, TWIST-1, PPARγ ".
3. Page 1 Line 20: The word "inflamed " should be changed to "inflammed".
4. Page 1 Lines 41-43: The sentence "Failure to recruit new adipocytes, due to dysregulation of adipogenesis (in both the proliferation and differentiation processes) leads to hypertrophic adipocytes with degradation and deposition of new extracellular matrix (ECM)" has no references. Please, add one.
5. The novel points in this study should be clarified because there are previous studies that discussed a similar topic.
6. Page 2 Line 76: "Adipose cell" should be replaced with "Adipocytes".
7. A collective diagram that summarizes the findings of the present study should be added.
8. Discussion: The role of PPAR gamma and irisin in adipose tissue dysfunction should be discussed in a more detailed manner.
9. The conclusion should include the possible clinical applications of the results of the present study.
10. The "Conclusion" section should be written at the end of the manuscript before "References" section.
11. Page 10 Line 335: A reference for the equation that determines the proliferation index should be added.
12. The catalog numbers of the used kits and chemicals should be added.
13. Page 12 Line 380: "Elisa" should be changed to "ELISA".
14. References should be written according to the format of Int. J. Mol. Sci.
15. The manuscript should be checked regarding the grammatical and typing errors.
Moderate English editing is required
Reviewer 2 Report
In the manuscript titled "Extracellular matrix remodeling induced by TNF-α is an early key event in adipose tissue dysfunction via NF-kB cell signaling in an MMP-dependent manner" by Laura Tomasello and colleagues. They have reported that in inflamed adipose cells a TIMP-1/TWIST-1 imbalance leads to a drop of PPARγ gene expression. The overactivation of MMP-9 leads to a dramatic impairment in adipogenesis and lipid accumulation ability. The presence of inflammation is associated with increased levels of adipose secreting irisin in early adipose commitment cells and increased levels of circulating irisin in patients with visceral obesity. Regarding the present manuscript, I would like to make a few remarks.
-The introduction should emphasize more the results (pros and cons) regarding irisin, as well as the fact that more than just single cells were used in the study
-It may be appropriate to place this section of the population under the ethical section. It is important to know this information at the beginning of the material and methods section in order to understand the study.
-The study may require a schematic representation of how it was conducted
-Were are defined these cell lines, ASC52Telo (line1), ASC52Telo 120 + TNF-α(72hrs); SATob, SATobTNF-α(72hrs)
-Is there anything significant in Figure 4?
-Figure 5 presents the results of the current study in a clear and concise manner
-The main weakness of the present study is the definition of the different types of tissues employed by the authors. There is a need for more details to highlight the novel findings. There are many key genes that can be identified easily and may help the authors to gain a better understanding of the overall picture. As a final point, the irisin needs to be discussed in a more comprehensive manner.
Reviewer 3 Report
The current study looked at the role of Tumor Necrosis Factor-α (TNF-α) in extracellular matrix (ECM) remodeling as well as the possible role of irisin in response to adipose tissue (AT) impairment. I'd like to make the following observations.
1. In the introduction section, irisin levels are linked to the severity of inflammation, which appears to be more consistent with the reference(s).
2. Obesity and inflammation do not appear to be clearly linked in the introduction.
3. The levels of internal standard (Beta-Actin) in Figure 2's Western Blots need to be improved.
4. The role of brown and beige adipose tissue appears to be overlooked in the current study. Why?
5. Describe "irisin as a driver of browning" in detail.
6. The discussion should include inflammatory cytokine blockade and AT impairment.
7. The current report's limitation(s), including the small sample size in each, may strengthen it.
Round 2
Reviewer 2 Report
Thank you for taking into account my previous comments. Based on my previous comments, the changes to the manuscript appear to have improved the manuscript. My only concern is the quality of the schematic figures.
Author Response
Thanks for your advice.
The schematic figures 6b and 6c were been replaced with figures of improved quality